# Emerging Role for 7T MRI and Metabolic Imaging for Pancreatic and Liver Cancer

**DOI:** 10.3390/metabo12050409

**Published:** 2022-04-30

**Authors:** Debra Rivera

**Affiliations:** Electrical Engineering Department, Eindhoven University of Technology, P.O. Box 513, 5600 MB Eindhoven, The Netherlands; d.s.rivera@tue.nl

**Keywords:** cancer, magnetic resonance spectroscopy (MRS), lymph node ratio, treatment response, biomarkers, metabolomics, pancreatic cancer, liver cancer, in vivo metabolic imaging

## Abstract

Advances in magnet technologies have led to next generation 7T magnetic resonance scanners which can fit in the footprint and price point of conventional hospital scanners (1.5–3T). It is therefore worth asking if there is a role for 7T magnetic resonance imaging and spectroscopy for the treatment of solid tumor cancers. Herein, we survey the medical literature to evaluate the unmet clinical needs for patients with pancreatic and hepatic cancer, and the potential of ultra-high field proton imaging and phosphorus spectroscopy to fulfil those needs. We draw on clinical literature, preclinical data, nuclear magnetic resonance spectroscopic data of human derived samples, and the efforts to date with 7T imaging and phosphorus spectroscopy. At 7T, the imaging capabilities approach histological resolution. The spectral and spatial resolution enhancements at high field for phospholipid spectroscopy have the potential to reduce the number of exploratory surgeries due to tumor boundaries undefined at conventional field strengths. Phosphorus metabolic imaging at 7T magnetic field strength, is already a mainstay in preclinical models for molecular phenotyping, energetic status evaluation, dosimetry, and assessing treatment response for both pancreatic and liver cancers. Metabolic imaging of primary tumors and lymph nodes may provide powerful metrics to aid staging and treatment response. As tumor tissues contain extreme levels of phospholipid metabolites compared to the background signal, even spectroscopic volumes containing less than 50% tumor can be detected and/or monitored. Phosphorus spectroscopy allows non-invasive pH measurements, indicating hypoxia, as a predictor of patients likely to recur. We conclude that 7T multiparametric approaches that include metabolic imaging with phosphorus spectroscopy have the potential to meet the unmet needs of non-invasive location-specific treatment monitoring, lymph node staging, and the reduction in unnecessary surgeries for patients undergoing resections for pancreatic cancer. There is also potential for the use of 7T phosphorous spectra for the phenotyping of tumor subtypes and even early diagnosis (<2 mL). Whether or not 7T can be used for all patients within the next decade, the technology is likely to speed up the translation of new therapeutics.

## 1. Introduction

Despite the many causes of cancer, all cancerous lesions present with a hallmark metabolic shift to glycolysis [1]. The metabolic profile of a cell is downstream of a complex interaction of numerous factors including: cell type, life style, genetic expression, environment, and local tumor environment (Figure 1 adapted from [2]). It is for this reason that metabolomics derived from phosphorus nuclei at magnetic field strengths of 7 tesla (T) dominate preclinical drug development [2]. While magnetic resonance imaging (MRI) was invented to allow non-invasive localized metabolomic monitoring of tumors for diagnosis and treatment steering, it is only in recent years that the stable high magnetic-field-strength (>4 tesla) magnets large enough to accommodate a human, and methods for use, have become available for clinical use.

The genetic instability of hepatic and pancreatic tumors causes a deadly situation in which during the course of treatment, tumor-resistant strains can newly arise or be given more space to thrive [3,4]. Genetic instability leads to massive tumor heterogeneity [5] and correlates with poor prognosis [3]. Whereas, pancreatic adenocarcinomas and hepatocellular cancers are characterized by heterogeneity; prostate cancer, which has a low mutation rate, is associated with low mortality [6]; hypermutation occurs in less than 5% of patients with prostate cancer [6]. Likewise, high mutation burden is uncommon in breast cancer (5% of breast cancers) [7].

The molecular taxonomy of pancreatic ductal adenocarcinomas includes dozens of classes of genetic mutations, which can occur individually or in combination [4]. The diverse types of tumor lines make a single serum marker for pancreatic cancer unlikely. Existing single serum approaches have specificity and sensitivity below 80%, whereas serum panels of proteomic markers have sensitivities and specificity approaching and exceeding 90% [8]. Even serum panels, however, are incapable of distinguishing adenocarcinomas derived from pancreas as compared to other organs containing glandular cells, and they remain clinically indistinguishable prior to resection [9]. While there are initial promising indications for treatment selection based on biomarkers, the genetic instability and thus heterogeneous distribution of tumors within a patient presents a confounding challenge [10].

Currently, there is a great risk that the full heterogeneity of the tumor, and the evolution over time, will not be captured. This is due to the inability of biopsies to sample the entire tumor(s), as well as the averaging effects of quantitative biomarkers and assays [11]. Therefore, “we may risk expending large resources on the development of fundamentally flawed approaches to biomarker-directed therapeutics” [11]. The potential for dramatic improvements in treatment outcomes for hepatic and pancreatic primary tumors through proteomics and metabolomics, may require pairing proteomics panels with novel methods for monitoring of treatment response in a location-specific manner [11].

While there are candidate patient populations with elevated risk of developing pancreatic cancer, there is no consensus on how to monitor these patients [12]. The heterogeneous nature of liver and pancreatic tumors may be a contributing factor that makes imaging more difficult, including tumor boundary delineation [13]. There is insufficient sensitivity across existing modalities for detecting small pancreatic ductal adenocarcinomas (<2 cm): ultrasound 39%, CT 40%, MRI 24%, and endoscopic ultrasound (EUS) 56% [14]. Meta-analysis has shown that the imaging modalities (MRI, PET/CT, CT, and EUS) are not statistically different, with the exception of PET, which has been shown to be slightly less accurate [15]. Radiomics is the use of machine learning using multiple parameters and/or modalities to evaluate whether treatment response outperforms established single-parameter imaging approaches. While radiomics is more beneficial than single imaging parameters for establishing prognosis in pancreatic cancers, more development is needed to attain sufficient accuracy to warrant clinical use [13].

Approximately one in four patients undergoing resection for pancreatic cancer are found during surgery to be unresectable, due to being locally advanced or due to distant metastases [16]. The direct and indirect costs of such an exploratory surgery exceed USD 40,000 [17], not to mention the physical and mental burden to patients and their loved ones. Non-invasive lymph node staging [18], and repeatable measures of treatment monitoring of primary tumors [8,19] present additional unmet clinical needs.

This manuscript will draw on the literature in order to make the case for the emerging and future role of 7T magnetic resonance imaging and phosphorus spectroscopy in the fight against deadly solid tumors, focusing specifically on hepatic carcinomas and pancreatic ductal adenocarcinomas. This review is not a systematic review, but rather provides an interdisciplinary survey of the literature. The first aim is to formulate a list of unmet clinical needs in pancreatic and hepatic cancers. The second aim is to draw on the work to date in the fields of phosphorus spectroscopy preclinical magnetic resonance imaging and nuclear magnetic imaging (NMR), and recent advances in 7T phosphorus spectroscopy in humans in order to provide evidence for 7T phosphorus spectroscopy in meeting the unmet clinical needs. The final aim is to propose a feasible strategy for leveraging 7T mri and spectroscopy to address unmet clinical needs.

## 2. Prior to Treatment

### 2.1. Detection and Staging

The state of the art in pancreatic cancer detection and staging is insufficient (see Table 1).

A key limitation in the staging of solid tumors with high heterogeneity is the limited sensitivity for identifying small tumors. The diagnosis of hepatocellular carcinoma with MRI has sensitivity and specificity of 72–76% and 92% [22], whereas the respective numbers for pancreatic ductal adenocarcinoma are 88.5% and 63.4% [21]. Figure 2 is a plot derived from the Surveillance, Epidemiology, and End Results (SEERs) database for diverse cancers, suggesting a strong correlation between early diagnosis and 5-year survival (R^2^ = 0.8; tumor types were selected with the aim of providing distributed sampling across the 5-year survival percentages; once selected, the cancer type was added to the plot irrespective of the given value for early diagnosis). More liver cancers are detected at the early stage than pancreatic cancers (45% vs. 11% detected as locally confined primary tumors), and survival is better in patients with liver cancers as compared to pancreatic cancers (20.3% vs. 10.8% 5-year survival).

#### 2.1.1. Imaging

Imaging at 7T can combine sub-millimeter resolution dynamic contrast enhanced MRI (0.7 mm isotropic), diffusion weighted imaging (1 mm isotropic), and phosphorus chemical shift imaging of voxels 2 cm per side in less than an hour [23]. The 2 cm isotropic voxel at 7T in the liver [24] is a dramatic improvement over the typical 6 × 6 × 10 cm voxel used for phosphorus spectroscopy of the liver [25]. The combined sub-millimeter resolution proton imaging, dynamic contrast enhanced MRI, and diffusion weighted imaging has the potential to identify small tumors in the pancreas, liver, and peritoneal space—with greatly enhanced ability to resolve the boundaries of tumors and inform the degree of invasiveness. In a breast cancer study (*n* = 40), combined 7T multiparametric imaging with diffusion weighted imaging and dynamic contrast enhanced imaging resulted in complete agreement between the two readers, no false negatives and two false positives (5%), for a diagnostic accuracy of 95.7% (tumor size: range 6–95 mm, mean 23.3 mm) [26]. The authors determined that the use of 7T multiparametric MRI would have avoided 6 out of 8 unnecessary biopsies (*n* = 40) [26]. A similar study by the same group at 3T in breast cancer (*n* = 106) yielded diagnostic accuracy of 91–93% for a range of readers, with a range of 5–8 false positives (mean 6.75), and 2–3 false negatives [27].

For body imaging at 7T MRI, even an approximate design for a half-wave antenna can provide extraordinary results with impressive longitudinal coverage, making the lack of a full-body birdcage for proton imaging an easily surmountable issue (Figure 3). Furthermore, a sufficiently uniform excitation profile is easily managed by what is termed parallel transmit phase shimming [28]. Such methods are now routinely implemented with user interfaces that feature region-of-interest selection on the console. The liver can be fully visualized with either two antennas or the full cross-section of the human under study using, for example, eight antennas with parallel transmit (see Figure 3). The wavelength for phosphorous (31P) investigations at 7T is comparable to that of conventional proton imaging at 3T; therefore, a standard birdcage for excitation paired with a local receive array is possible [29,30].

#### 2.1.2. Spectroscopy

The Warburg effect predicts elevated ratios of phosphomonoesters relative to beta nucleoside-triphosphate (NTP), which can be amplified as much as 7-fold in progressing tumors and reversing in the case of responding tumors [31]. Mass spectroscopy of 31 tissue samples from patients with hepatocarcinoma cell tumors all presented with more phosphocholine than the adjacent tissue [32]. Cox et al. demonstrated that all the patient volunteers with primary liver tumor (*n* = 4) had more phosphomonoester to phosphodiester ratios (integral of the metabolite peak) than *each* of the healthy volunteers (*n* = 28) (Figure 4a) [33]. Brinkmann et al. found that even voxels with less than 50% tumor (24 total, metastatic liver, 8 with <50% tumor volume in voxel) were statistically different from the tissue from healthy volunteers [34]. In vivo phosphorus spectroscopy at 1.5 and 3 T lacks the spatial resolution for imaging in the pancreas. While proton spectroscopy is not able to discern phospholipid metabolites [35], it can be completed with fine enough spatial resolution at even 1.5 T in patients and provides indirect evidence for elevated phospholipids in human pancreatic tumors [36]. In a study with 40 patients, proton spectroscopy yielded a significantly higher (*p* < 0.05) ratio of choline to lipids in tumors compared to normal pancreas [37]. Phosphorus spectra of human-derived cell lines of pancreatic cancer show elevated phosphocholine and/or phosphoethanolamine, with unique fingerprints across cell lines [38,39]. A metabolomics analysis of tissue samples from patients (*n* = 106; 66 of whom received neoadjuvant chemotherapy) undergoing resection for pancreatic adenocarcinomas pinpointed increased levels of ethylene (*p* = 0.0078) and choline (*p* = 0.0014) as predictors of short-term survival [40]. While it is difficult to draw conclusions from the single time point metabolomic analysis for the samples with and without neoadjuvant treatments, the trends according to long- and short-term survival provide insight. Both of these metabolites feed into the Kennedy pathway which synthesizes phosphocholine and phosphoethanolamine. In a further study with 8 cell lines derived from patients with pancreatic adenocarcinoma, all expressed elevated phosphocholine [41]. Over-expression of the enzyme choline kinase alpha was found in 90% of the pancreatic tumors evaluated in tissue microarrays [41]. Choline kinase alpha is associated with both branches of the Kennedy pathway (phosphorylation of choline and ethanolamine). The overexpression of choline kinase alpha is associated with oncogenesis; therefore, the enzyme is a therapeutic target [42].

In one of the first 7T studies of breast cancer, Wijnen et al. found that the concentration of phosphocholine and phosphoethanolamine in three patients was greater than that of the 11 healthy volunteers (Figure 4b) [43]. There is some indication that 7T MRI and spectroscopy are more sensitive to small tumors. A recent study of 50 breast cancer patients demonstrated that the need for systemic therapy post-operatively could be predicted by ultra-high field imaging at 7T, including diffusion weighted imaging and phosphorus spectroscopy, but only for patients with small tumors (<2 cm) [23]. This is unsurprising, due to the tendency towards Gomperzian growth in solid tumors, in which (like logistic growth curves) the most rapid rate of growth occurs before the tumor outgrows the ability of the vasculature to provide oxygen and nutrients [31]. The conclusion from the 50 patient dataset from Schmitz et al., in which the phosphomonoesters were most predictive for tumors less than 2 cm [23], supports the hypothesis that smaller tumors have higher concentrations of phosphomonoesters—as did the demonstration of high signal to noise ratios for the individual phosphomonoester peaks in *sub-centimeter* tumors with histologically validated high rate of cell division (same dataset as [23]). As mentioned above, unlike proton spectroscopy [44], previous in vivo studies suggest that phosphorus spectroscopy can be accurately characterized even when the tumor makes up less than half of the voxel [34]. Phosphorus spectra from a healthy sub-centimeter lymph node has also been obtained despite less than 0.5 mL volume [45]. The evidence, therefore, warrants consideration for use of the method even for tumors in the of 1–2 mL range, if not smaller.

Recently, we obtained the first 7T phosphorus spectra of an advanced metastatic liver cancer (Figure 5) [24]. The imaging was conducted using antennas and B1 shimming for proton imaging and two loops tuned to the phosphorus resonance frequency (120.6 MHz) for collecting spectra. The phosphoethanolamine (PE) of tumors (Figure 5b) is greater than the phosphocholine (PC), glycerophosphoethanolamine (GPE), glycerophosphocholine (GPC), and inorganic phosphate (Pi), while the peak height of PE is less than or equal to those of PC, GPE, GPC, and Pi in the liver voxel (Figure 5c). As can be seen from a healthy volunteer, in the healthy liver (Figure 5d), the phosphodiesters (GPC and GPE) dominate the phosphomonoesters (PE and PC). We conservatively estimated that there is sufficient signal sensitivity to detect and/or monitor liver cancers as small as 2 mL. The conservative nature of the estimate is rooted in the false assumption that the late-stage tumor has the growth rate and, therefore, phosphomonoester concentrations of a tumor undergoing rapid growth. Therefore, it is likely, as suggested in the breast cancer data discussed above, that the method is sensitive to tumors smaller than 2 mL.

A group in the Netherlands has begun imaging patients with pancreatic cancers with 7T scanners with an intention of using phosphorus spectra to provide a non-invasive and non-radiative modality for staging neoadjuvant treatment prior to resection. The proof-of-principal establishes reproducibility of measurements of metabolite concentrations in healthy volunteers and demonstrates the first 7T phosphorus spectra from a primary tumor (94 × 41 mm; 2 cm isotropic voxels) in a patient with pancreatic ductal adenocarcinoma [46]; the larger clinical study is underway.

### 2.2. Lymph Node Imaging and Virtual Biopsy

The lymph node ratio is the proportion of dissected lymph nodes that are positive. For the majority of solid tumors, lymph node ratio is the most important prognostic factor for long-term survival after surgery [46,47,48]. In a metadata analysis of patients undergoing resection surgeries for pancreatic adenocarcinomas (*n* = 197), positive lymph nodes were found in 75.6% (range 56–83%) [16]. Lymph node size is not a predictor of involvement [18]. Even for breast cancers, the state-of-the-art for lymph node staging remains invasive with considerable morbidity; however, radiomics approaches analyzing characteristics of the primary tumor to predict lymph node status are an active area of research [49]. With pancreatic cancers and other solid tumor types, staging is possible only after lymph node dissection [50].

Recent efforts at 7T suggest that the technology can support direct non-invasive lymph node staging [23,45]. The enhanced signal-to-noise ratio of ultra-high field MRI allows imaging of even sub-centimeter lymph nodes. For example, in the axilla, it was possible to image lymph nodes with maximum diameter of 3 mm [45]. Phosphorus spectroscopy (1.3 cm isotropic voxels) from a single healthy axilla lymph node (3 mm minor axis; 8 mm major axis) allowed detection of <2 mM concentrations of phosphocholine [45].

### 2.3. Aggressiveness and Tumor Sub-Typing

An elevated concentration of the phosphodiester glycophosphocholine (GPC) is correlated with increased cell migration capabilities and likewise prevalence of invasiveness and metastasis [51]. We have observed elevated GPC in a metastatic liver patient at 7T [24], and we have seen correlations with high mitotic count (histological marker of aggressiveness; number of cells in mitosis per 2 mm^2^) in small tumors (sub-centimeter) with the elevated GPC spectra in the datasets from [23], including in a single lymph node.

It has been speculated that in the context of the wide mutational variation present in pancreatic ductal adenocarcinoma, classifications based on molecular subtypes will lead to improvements over imaging approaches for prognosis, risk stratification, tailored treatments, and clinical trial patient selection [4]. Breast cancer cell lines, for example, exhibit phosphorus spectroscopy metabolic fingerprints [52]. Phosphorus spectroscopy, in animal models, is useful in differentiating healthy versus tumor pancreatic tissue, and also shows variations in metabolic ratios for different tumor types and models [38]. Likewise, phosphorous spectroscopy has been used for decades (e.g., [53]) to cross validate between different preclinical models for hepatic tumors. Phosphorus spectroscopy can also be used to inform the tumor microenvironment, e.g., pH, hypoxia, and can give an indication of what percentage of the tumor is well perfused [31]. Specifically in pancreatic ductal adenocarcinoma, stellate cells produce more proteins that cause rigidification of the surrounding tissue and blood vessel collapse, impairing drug delivery and creating more extreme hypoxia that further promotes tumor migration and resistance to therapeutics [10]. Hypoxia, as assessed by phosphorus spectra of the mucosal lining of patients with gastric cancer, showed that overall survival was worse in those with pre-morphologic changes in tissue surrounding the tumors indicating mild and severe hypoxia [54]. For patients with soft tissue sarcomas, the pretreatment phosphomonoester to phosphodiester ratio correlated strongly with metastasis-free survival [55]. The role of hypoxia is a known factor in pancreatic cancers leading to poor prognosis [56]. The interplay of tumor energetic status and the modulation thereof by treatment (see Figure 6 from [57]) are critical for both improving treatment efficacy and interpreting treatment response. Phosphorus spectroscopy can inform: proliferation, energetic status, and hypoxia, while MR imaging can inform tumor size, vasculature morphological changes, and blood supply/perfusion. The complex dynamics can be monitored with phosphorus spectroscopy.

## 3. Treatment Response

### 3.1. Need and State-of-Art

Biomarkers that provide guidance on treatment selection, and that can inform treatment response are greatly needed—especially in place of costly treatments in terms of patient burden and/or financial cost when they are not effective [8]. For patients with resectable pancreatic cancer, progression-free survival is on average 13 months while overall survival is on average 25–28 months in patients with pancreatic ductal adenocarcinomas; and for patients, unresectable cancer median progression-free survival is 3.3–6.4 months and overall survival is 7–11 months [58]. In consideration of both the short time frame for survival and the heterogeneous cell line pool supporting the evolution of drug resistance, there is a great need for non-invasive, location-specific methods for monitoring early treatment response.

The difficulty of monitoring treatment response with structural images is a major hindrance to precision medicine methods for treating pancreatic ductal adenocarcinoma [59]. The MRI-derived apparent diffusion coefficient—one of the most promising imaging modalities for quantifying treatment response—outperforms typical imaging criteria for tumor response (e.g., RECIST and mRECIST) though is still insufficient for distinguishing between responders and non-responders to chemotherapy [60]. Multi Detector CT, and other imaging modalities, suffer from an inability to distinguish between infiltrating tumor and fibrotic tissue, and therefore there is an immediate need for new non-invasive methods for evaluating response to chemotherapy to support re-staging after neoadjuvant therapy [61]. The most conclusive evaluation, therefore, remains histopathology, which is available only to the small fraction of patients who (a) qualify for treatment and (b) undergo therapy prior to surgery. Patients receiving chemotherapy prior to surgery can be evaluated for treatment response surgically, but the one time point on offer is at the earliest, after 14 weeks [62].

The initial patient studies for patient-derived organoid prediction of response, termed pharmaco-phenotyping, have yielded success rates of 33–100% for liver cancers and 60–88% for pancreatic ductal adenocarcinomas [58]. While there is a risk of sampling bias inherent to biopsy, patient-derived organoids, can provide insight into predicting a good candidate treatment, but the rapid establishment for such a model is 6 weeks [58]. A subsequent patient-derived organoid culture would take another 6 weeks—discounting any needed time for applying a treatment—again, taking the patient beyond a three-month period which is the beginning range for continuance in patients with non-resectable pancreatic adenocarcinoma. This is likely an approach for treatment development and gaining insight to tumor subtypes that are candidates for treatments, rather than treatment monitoring.

### 3.2. Evidence Suggesting the Potential of Phosphorus Spectroscopy for Treatment Monitoring

In preclinical pancreatic cancer cell lines, phosphorus spectroscopy can distinguish between responders and non-responders [38]. Alterations in metabolite peaks (including the phosphomonoesters and phosphodiesters) involved in the Warburg effect evolve in vitro within 1 h of administration of chemotherapy [63]. Meyerhoff et al. used phosphorus spectroscopy at 2 T to study response to chemotherapy and embolization (blocking of blood supply), concluding that phosphorus spectroscopy is a means of directly monitoring metabolic changes in response to therapy for patients with hepatic tumors (*n* = 5) [64]. Phospholipid spectroscopy at 7T is useful preclinically for monitoring the effects of molecular target therapies, as well as identifying new targets [35].

Phosphorus spectroscopy is also adept at identifying responders for antiangiogenic treatments as well as clarifying the mechanisms of action (e.g., for patients with gliomas [65]). In breast cancer, 7T phosphorus spectroscopy accurately identified a histologically confirmed non-responder at all time points, whereas tumor shrinkage and proton spectroscopy initially indicated that the patient was a responder [66]. Similarly, results from another cohort of patients with breast cancers volunteering for 7T phosphorus spectroscopy, indicated that the ratios of phosphocholine and phosphoethanolamine to their respective glycosylated counterparts went up after the first round of chemo for the non-responders, down somewhat for partial responders, and down the most in complete responders [67].

In solid tumors, in addition to modulation of the phospholipid metabolism via the Warburg effect, metabolic changes (nucleotriphosphate, phosphocreatine, and internal pH) associated with the percentage of surface area of well-perfused tumor can also inform treatment effects [31], including conditions in which tumors previously starved for nutrients can again thrive. In animal models, phosphorus spectroscopy of hepatic treatment response includes reduction of phosphomonoesters, pH changes, responses to changes in the microenvironment, and availability of nutrients [53]. While pH measurements derived from the inorganic phosphate peak are typically interpreted as indications of the internal cellular pH, a recent study monitoring changes due to hyperthermia in a tumor model along with independent internal and external pH measures indicated that the observed inorganic phosphate peak splitting led to a lower resonance that was more in accordance with the average internal pH (7.48), while the higher resonance peak was more in accordance with the external pH (7.14) [68]. Early in vivo exploration of phosphorus spectroscopy for monitoring of patient response to chemotherapy by Redmond et al. found that across diverse large soft tumors (e.g., lymphoma, breast, adenocarcinoma of the neck; *n* = 16, 2 of 16 where non-responders) consistently non-responders had an increase in internal pH post therapy, yet partial responders and complete responders had a decrease in internal pH post therapy [69].

## 4. Dosimetry and Treatment Development

Naruse et al. demonstrated a dose-dependent response, that in 10 of 10 animals (rats inoculated with rat glioma cells) given the highest hyperthermia dose (5 watts continuous, 60 min) elevated inorganic phosphate peak and decreased nucleotide triphosphate peak within the first hour of heat application predicted and preceded the histological response (necrosis), that was observable in imaging two days later [70]. None of the animals receiving doses less than 3 watts exhibit changes in either phosphorus spectra or imaging, but those animals receiving between 3 and 5 watts had mixed responses in which phosphorus spectroscopy accurately predicted histological response [70]. Similarly, James et al. (see Figure 7) found that the ratio of the nucleotide triphosphate to inorganic peak height was able to track a response to the hypothermia during heat application [71]. Kaplan et al. used phosphorus spectroscopy for continual monitoring of diverse preclinical pancreatic ductal adenocarcinoma models during cytotoxicity studies to identify chemo therapeutic agents and combinations thereof for otherwise chemo-insensitive cell lines based on the phospholipid metabolites, with pH and high energy phosphate peaks allowing for elucidation of the mechanisms of actions of the compounds [38]. Stijens et al. showed a temperature-dependent response to hyperthermia in an animal model, and even further demonstration of the percentage of the metabolite changes when combined with radiotherapy; furthermore, early changes (15 min) were indicative of changes in tumor perfusion while later changes (24 h) were predictive of future necrosis [72]. Thermal sensitivity to treatment is also modulated by the internal pH, as has been shown by phosphorus spectroscopy [54]. Based on the study in small cell lung cancer xenographs in mice, Kristjansen et al. propose that ATP/Pi ratio as evaluated with phosphorus spectroscopy is likely of value for early response dosimetry and gauging radiosensitivity of tumors [73]. In a follow-up study, Kristjansen et al. examined radiation doses of 2.5, 10, and 40 dose (Gy) in mouse brain and two human-derived small cell lung tumor xenographic models, finding alterations in ATP/Pi only for the tumor cell lines and only with the 40 Gy doses [74]. The ATP/Pi ratio can also be used to determine not only the optimal dose, but also the optimal schedule of tumor irradiation (model: hypoxic murine mammary carcinoma) [75]. No studies of dosimetry in patients were found by the author.

Molecular targets have been identified and explored through analysis of phospholipid metabolism. A study of the choline metabolite profiles has led to the identification, for example, of choline kinase alpha as a molecular target [76]. A choline kinase alpha inhibitor (MN58b) is now a molecular treatment for pancreatic ductal adenocarcinomas; gemcitabine-resistant pancreatic tumor cells displayed enhanced sensitivity to CHKa inhibition and, in vitro, MN58b improved the effects of three main chemotherapies tested (gemcitabine, 5-fluorouracil, and oxaliplatin) [77].

## 5. Perspective

As preclinical models do not accurately match conditions in patients, as written by Kaplan in the late 1990s, the clinical role of in vivo phosphorus spectroscopy will ultimately be determined in humans [38]. Bell et al. conducted a late 1990s review of the clinical literature for phosphorus spectroscopy in treatment monitoring for patients with hepatic tumors at conventional field strengths, and found that the method was effective, but with the primary limiting factors of a lack of spatial resolution and the inability to distinguish the contributing peaks for the critical phospholipid metabolites [78]. Two decades later, thanks to the efforts of the researchers and engineers of the growing ultra-high-field community, we are prepared to pick up the work in vivo.

The critical clinical issues are numerous. Tumor heterogeneity is a confounding variable for: biomarker assays, tumor imaging and delineation, biopsies for tailored treatments, and response to a given treatment. There is a great challenge in identifying tumors 0.5–2 cm, for early diagnosis, and also proper staging. The powerful prognostic markers of lymph node positive number and lymph node ratio can be assessed only via surgery. Microvasculature invasion (particularly for hepatocellular carcinoma) is difficult to identify non-invasively. Clinically derived metrics for treatment monitoring are lacking. Radiomics, while better than single-parameter imaging, is still insufficient for clinical use. Biopsies may miss non-responding components; likewise with assays. Neoadjuvant treatment complicates already insufficient imaging-based treatment assessment, and surgical assessment is not always available and only allows for one time point. There is a considerable burden for evaluating new therapeutics. Tumor heterogeneity complicates identifying whether a treatment works on a subtype of tumor, due in part to the difficulty in identifying a homogeneous patient population—or even a homogeneous tumor. The lack of non-invasive and localized treatment monitoring makes it difficult to identify sub-types of tumor cells that are responding to a treatment.

Through 7T imaging and spectroscopy, considerable possibilities become available. The enhanced signal to noise ratio at high field allows for improved spatial and/or temporal resolution for more reliable imaging of small tumors, to aid diagnosis and staging. Metabolic imaging opens up the possibility for: (a) confirming standard imaging results; (b) non-invasive assessment of positive lymph node ratio without relying on size; (c) informing the tumor microenvironment (e.g., hypoxia); (d) early and local response to treatment—identifying regions that are responding and those that are not; (e) metabolic phenotyping; (f) biomarkers for cross-comparison of tumor models to patient tumors; and (g) dosimetry on an individual level.

In consideration of the unmet clinical needs and the potential role for ultra-high field phosphorus spectroscopy and microscopy, there is sufficient justification to begin exploring the immediate benefit of 7T imaging and phosphorus spectroscopy with regards to staging. Given that 25% of patients undergoing pancreatic cancer resection surgeries are found to be inadequately staged, if the 7T scan costs an order of magnitude less than the surgery and finds half or more of the otherwise surgically discovered non-operable patients, the cost benefit analysis would suggest a benefit from scanning presurgically *all* patients who are candidates for surgery. One hour of scan time, for example at Scannexus in Maastricht, costs EUR 500. Including a budget for contrast agent, trained technicians, medical staff, and radiologists can reasonably fit within a USD 4000 budget. It is worthwhile, therefore, to immediately evaluate what percentage of unresectable surgeries can be identified through multiparameter 7T studies that combine conventional structural imaging with phosphorus spectroscopy, elastography, diffusion weighted imaging, and dynamic contrast imaging.

The overarching goal is to improve outcomes and reduce suffering for the most lethal cancers. Surgical resection is the only cure for patients with pancreatic ductal adenocarcinomas. Therefore any technology that can help ensure that those patients likely to benefit—and only those patients—receive curative surgeries, would be of great value. The potential for phosphorus spectroscopy to aid in early diagnosis for cancers has hitherto been out of the question due to the limited spatial resolution at lower field strength. For pancreatic cancers, and also liver cancers, existing imaging methods are insufficient for accurately identifying tumors in the 1–2 cm range. Early diagnosis is defined as tumors exhibiting a maximum dimension of less than 2 cm. Aberrant phospholipid metabolism distinguishes primary cancers from healthy tissue in the liver and pancreas. At high field, there is potential to do so for tumors 0.5–2 cm in size. Tumors in the rapid growth state (typically below 2 cm) and aggressive tumors have more extreme alterations in phospholipid metabolism, as well as low background noise. This allows for identification of tumors that make up less than half of the voxel volume. Therefore, it is worth asking if there is a role for monitoring patients at elevated risk for liver and hepatic cancers, including those indicated by serum panels. Through early detection, more patients can be diagnosed at a stage in which surgical resection is an option.

One of the greatest barriers to the development of new treatments is the inability to monitor treatment response non-invasively, over time, and according to location. With pancreatic and hepatic cancers, the short median progression-free survival times (6 for locally advanced; 13 months) increases the importance of being able to quickly identify whether a treatment is of benefit. Phosphorus spectroscopy at 7T has the potential to transform the therapeutic development pipeline from the current approach (one treatment on many patients), such that we can test many treatments (and if need be doses) on a single patient, as well as creating dynamic treatment approaches that empower clinicians to identify treatment resistance as it evolves and adapt accordingly.

Just as preclinical models are identifiable by the metabolic fingerprints of phosphorus metabolites (which requires differentiation of the sub-peaks of phosphomonoesters PE and PC), it is conceivable that 7T phosphorus spectroscopy will ultimately allow identification of dominant cancer subtypes in vivo.

The potential cost benefit of 7T imaging and spectroscopy for presurgical staging in patients with pancreatic ductal adenocarcinoma, presents a feasible pathway for evaluating further roles for both. For example, upon validating an approach within the context of pancreatic cancers, similar pipelines can be initiated for hepati c tumors and beyond (e.g., earlier diagnosis in ovarian cancers, and lymph node staging for breast cancers). Here is a proposed outline of steps to reduce suffering for patients with pancreatic cancers:Step (1) Evaluate what percentage of patients deemed in surgery to be non-operable can be identified preoperatively using 7T spectroscopy and imaging.Step (2) Reduce unwanted surgeries by informing infiltrating/non-infiltrating, liver status, and LN ratio. Use tumor microenvironment metric of pH to aid decision for identifying hypoxic tumors that are at high risk for recurrence. Test a subset of patients that are not surgical candidates post-therapy to see if treatment response is reliably measured by phosphorus spectra.Step (3) Evaluate phosphorus spectroscopy for guiding and adapting treatment in an individual.Step (4) Collect database for pairing metabolic subtypes with successful treatments, ensuring that heterogeneity within tumors is a guiding factor and not a confounding one (identifying proportion of population of healthy and malignant cell types based on metabolomics).Step (5) Search for low-cost biomarkers that are effective and can be collected non-invasively.

Metabolic profile and vascularization modulate the tumor microenvironment and the internal heterogeneity, which are in turn contributing factors to metabolomic spectra. For example, the concentration of phosphomonoesthers in an aggressive tumor depend heavily on how well the tumor is perfused. Therefore, multiparametric approaches are needed to ensure that tumor perfusion does not become a confounding variable. Tumor heterogeneity across patients and within even small tumors, complicates current “homogeneous”—population clinical trials, though, will be an asset for identifying commonalities for treatment-success aggregated patient cohorts. The combined technological advancements in the last decade of ultra-high field magnetic resonance scanners and multiparametric approaches could be fundamental to unlocking the potential of personalized and precision medicine. It is critical as well to contribute patient data to the Human Metabolome Database [79] in order to facilitate the ability to capture emerging patterns and phenotypes. Once metabolic phenotypes are characterized, it is worth searching for features from low-cost techniques, such as genomics and proteomics, as well as the use of micro-bubble contrast enhanced ultrasound for characterizing vasculature characteristics and kinetics [80], ultrasound elastography [81], and/or synthetic approximations of 7T results from lower field strength data [82] as applied to phosphorus spectroscopy. While it may be impossible to find low-cost surrogates for non-invasive tracking of metabolic changes for treatment steering, it is conceivable that a diverse array of subtypes of hepatocellular carcinomas and pancreatic ductal adenocarcinomas can be uniquely identified and paired with effective treatment protocols. The aforementioned work predicting lymph node involvement based on primary tumor features is a hopeful example of the power of texture analysis feature extraction [49].

## 6. Conclusions

In order to improve outcomes and reduce suffering in the often lethal pancreatic and hepatic cancers, there is an immediate need for improved non-invasive staging and diagnosis techniques. Furthermore, methods for monitoring treatment response non-invasively, and according to position within the body, stand to benefit both the development and implementation of personalized medicine. The preclinical literature supports the potential of ultra-high field (7T) magnetic resonance imaging and phosphorus spectroscopy to meet these unmet challenges, including noninvasive lymph node staging, dosimetry, pre-treatment risk stratification, and *early* treatment response. There may even be a role for diagnosis (0.5–2 mL tumors). It is time to analyze the use of 7T MRI and phosphorus spectroscopy for a cost benefit analysis in order to reduce futile surgeries for patients with seemingly resectable pancreatic adenocarcinomas, as a first step towards developing ultra-high field magnetic resonance into a platform for treatment development for deadly cancers.

## Figures and Tables

**Figure 1 metabolites-12-00409-f001:**
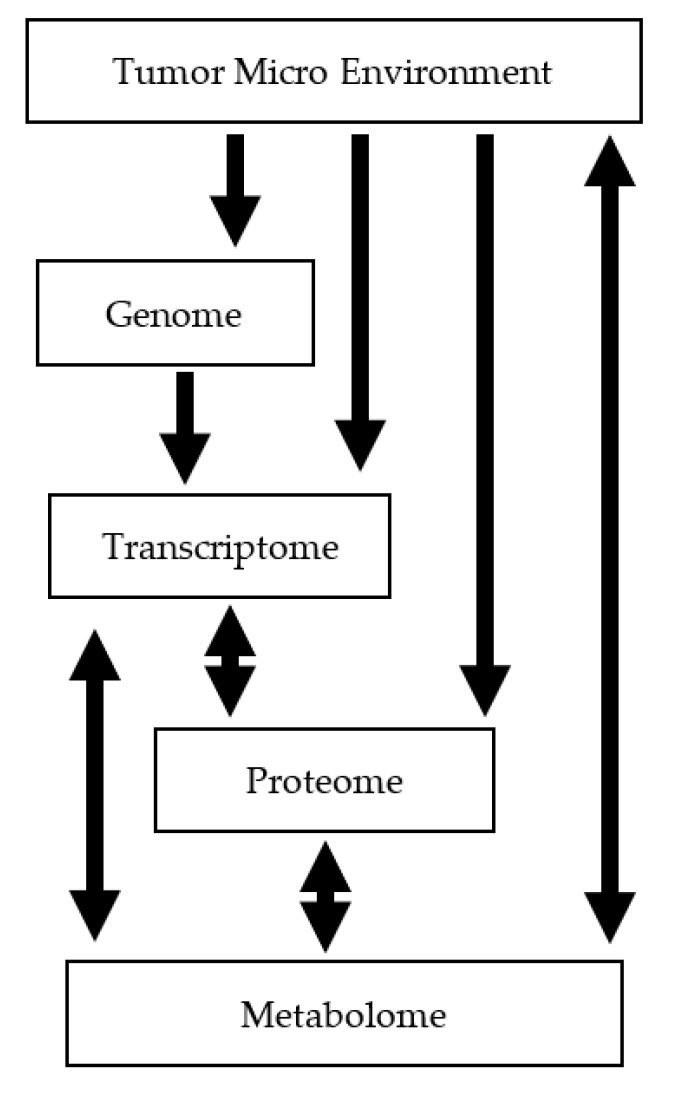
The metabolomic profile, or ‘Metabolome,’ is downstream of all other factors. Arrows indicate reciprocal/unidirectional modulation of factors. Adopted from Griffen et al. [2].

**Figure 2 metabolites-12-00409-f002:**
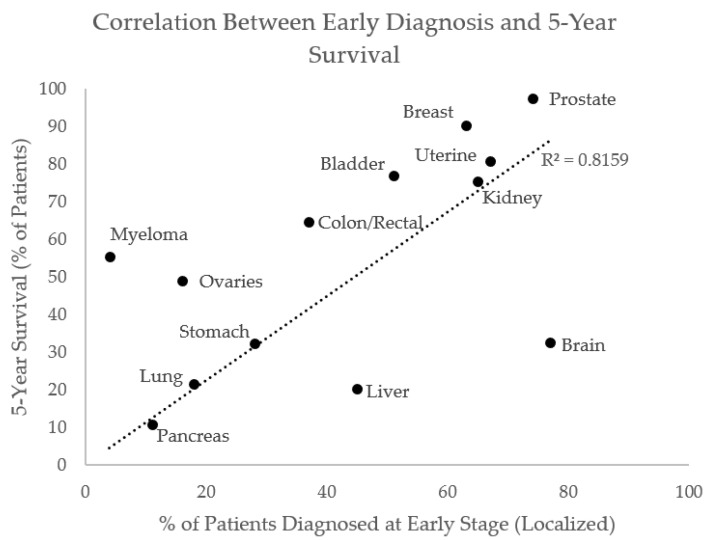
There is a strong correlation between early diagnosis and 5-year survival across cancer types. Data from: Surveillance, Epidemiology, and End Results (SEER) Program Populations (2011–2017) (www.seer.cancer.gov/popdata) accessed on 2 February 2022, National Cancer Institute, DCCPS, Surveillance Research Program, released February 2021.

**Figure 3 metabolites-12-00409-f003:**
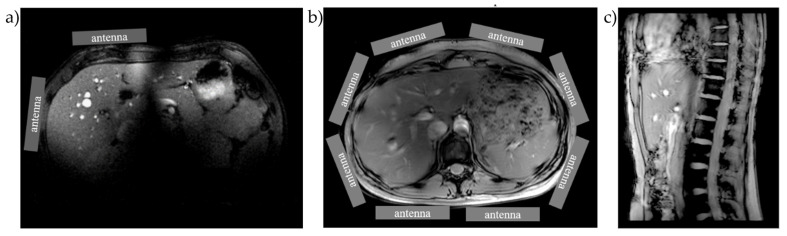
The capabilities of multiple antenna elements deployed for signal excitation and collection for proton imaging. (**a**) Two antennas (in quadrature) are sufficient for localizing the liver for phosphorus spectroscopic imaging; (**b**,**c**) clinical-grade proton imaging can be achieved with 8 antennas with parallel transmit and B1 shimming. In (**b**) the full extent of the liver is visualized in an axial slice (as well as white and grey matter in the spinal cord). In (**c**) a sagittal slice shows the longitudinal coverage provided by antennas, which will be useful for identifying metastases in the peritoneum. Images adapted from Rivera et al. [24].

**Figure 4 metabolites-12-00409-f004:**
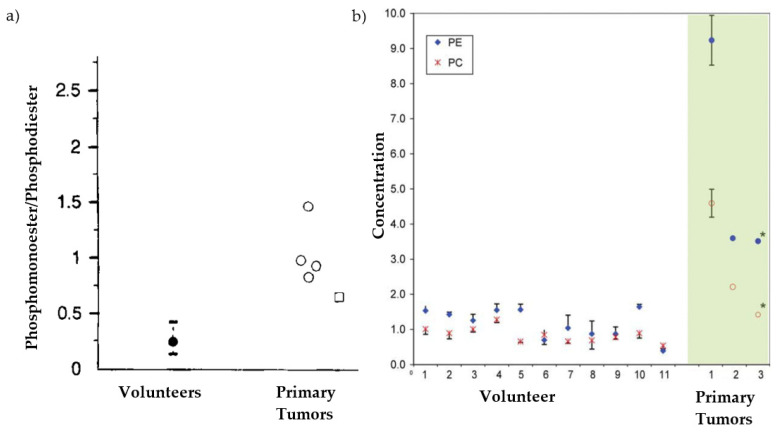
Non-overlapping measures for patients versus healthy tissue in vivo. (**a**) Data from primary liver tumors demonstrates that the phosphomonoester (e.g., phosphocholine (PC) and phosphoethanolamine (PE)) to phosphodiester ratio of the primary tumor spectra did not overlap with the ratios from any of the volunteers (*n* = 28; mean +/− 2 standard deviations); from Cox et al. [33]. Open circles represent hepatocellular carcinomas, while open squares indicate other primary liver tumors. Black circle indicates spectra from a healthy volunteer. (**b**) Similarly, the phosphomonoester (PE and PC) concentrations were non-overlapping with those of the healthy volunteers (7T, breast cancer; from Wijnen et al. [43]). Green shading delineates data points from tumors, and asterisks indicate datapoints collected using 1D chemical shift imaging instead of 2D.

**Figure 5 metabolites-12-00409-f005:**
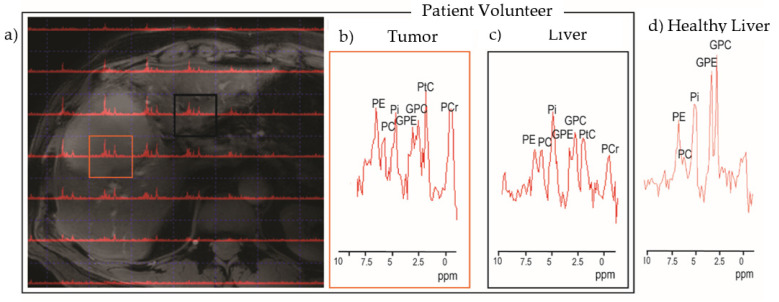
Images and phosphorus spectroscopy obtained at 7T field strength from healthy and patient volunteers. (**a**) Phosphorus spectra 3D chemical shift imaging with 30 mm isotropic voxels from a patient with liver metastasis (gastric primary), displayed on a B1-shimmed Dixon image obtained with eight antennas. Red square and black square correspond to tumor and liver tissue respectively. Spectroscopic data from a patient with one average (**b**,**c**) and from a healthy volunteer (**d**) with five averages, scaled to the Pi peak. PE, phosphoethanolamine; PC, phosphocholine; Pi, inorganic phosphate; GPE, glycosylated PE; GPC, glycosylated PC; PtC, phosphotidylcholine, PCr, phosphocreatine. Note that the chemical shift is expressed as parts per million (ppm) shift relative to phosphocreatine, which is circa 120.6 MHz at 7T. Images adapted from Rivera et al. [24].

**Figure 6 metabolites-12-00409-f006:**
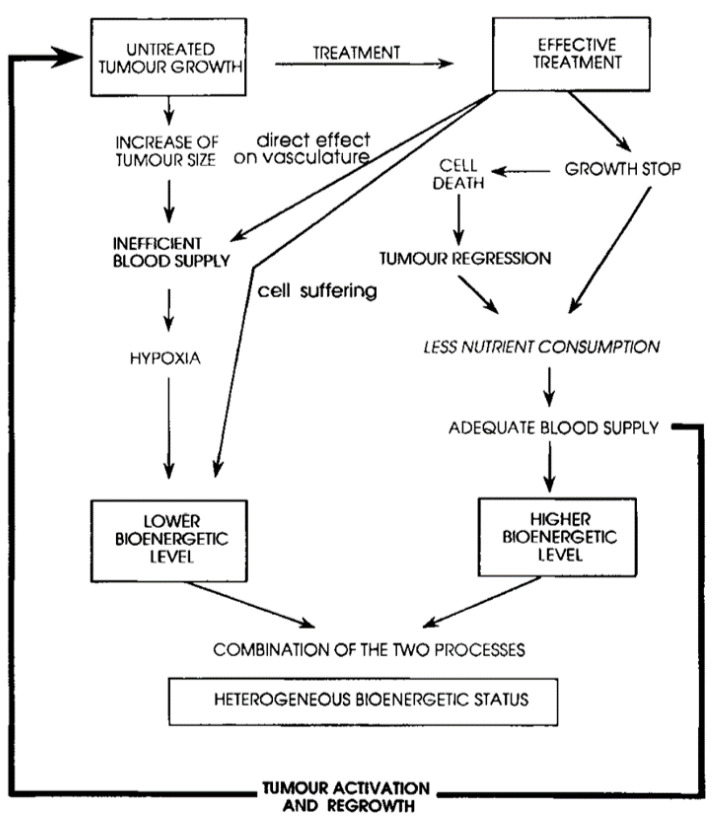
The various energetic pathways in proliferating and treated solid tumors. Image from de Certaines et al. [58].

**Figure 7 metabolites-12-00409-f007:**
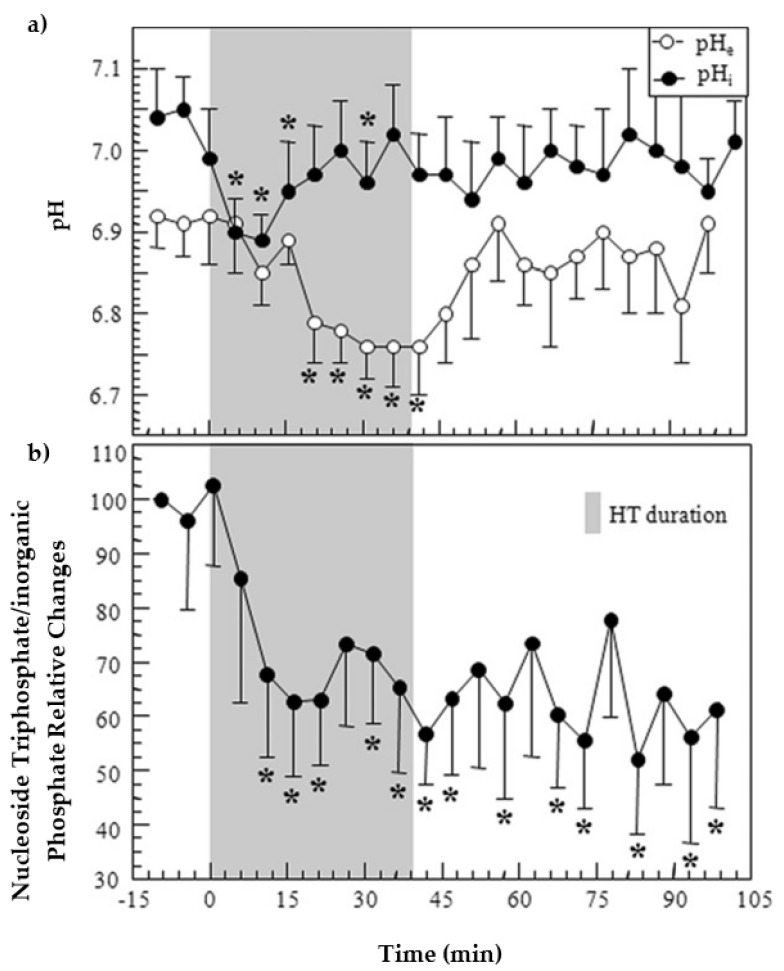
The changes in response to hyperthermia (HT). (**a**) The intracellular (pHi) and extracellular (pHe) pH changes in response to 45 °C hyperthermia. (**b**) The ratio of nucleoside triphosphate (NTP) to inorganic phosphate (Pi) as normalized to the baseline ratio. Please note: in vivo measurements from subcutaneously implanted 9L-gliosarcoma in Fisher rats (*n* = 6), asterisk (*) indicates *p* less than or equal to 0.05. Image from James et al. [72].

**Table 1 metabolites-12-00409-t001:** State of the art in pancreatic cancer. Abbreviations: computed tomography (CT); endoscopic ultrasound (EUS); magnetic resonance imaging (MRI) at 1.5 or 3 tesla. Data from Shrikhande et al. [20] and Costache et al. [21].

Author	Year	Factor Analyzed	Modality	Sensitivity	Specificity	Accuracy	Comments
Costache et al.	2017	Diagnosis	Helical CT	81%	43%	83%	EUS for detection; CT for determining resectability
			EUS	97%	90%	93%	
			MRI	88%	63%	89%	
Soriano et al.	2004	Locoregional extension	Helical CT	66%	100%	74%	Helical CT and EUS—most useful individual imaging techniques in the staging of pancreatic cancer
			EUS	44%	100%	62%	
			MRI	53%	100%	68%	
		Nodal staging	Helical CT	37%	79%	62%	
			EUS	36%	87%	65%	
			MRI	15%	93%	61%	
		Vascular invasion	Helical CT	67%	94%	83%	In potentially resectable tumors—sequential approach: initially helical CT followed by confirmatory EUS—most reliable and cost effective
			EUS	42%	97%	76%	
			MRI	59%	84%	74%	
		Distant metastases	Helical CT	55%	96%	88%	
			EUS	0%	100%	85%	
			MRI	30%	95%	83%	

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
