# Peer review of "Emerging Role for 7T MRI and Metabolic Imaging for Pancreatic and Liver Cancer"

_metabolites, 2022, doi:10.3390/metabo12050409_

Round 1

Reviewer 1 Report

In this manuscript “ Emerging role for 7 T MRI and metabolic imaging for pancreatic and liver cancer “  Debra Rivera presents a recent approach to analyze the use of 7 T MRI and phosphorus spectroscopy for a cost benefit analysis in order to reduce unnecessary surgeries for patients with seemingly resectable pancreatic adenocarcinomas, as a first step towards developing ultra-high field magnetic resonance into a platform for treatment development for deadly cancers.

The manuscript is well discussed and structured, however the objectives of the review are unclear and lacks on specificity. This is a manuscript heavily based on earlier works as none of the figures are original

The abstract is also too vague being difficult to understand the main aim od the work and the procedures followed to write the manuscript. Research from PubMed in last years? The design followed must be described.

In all Figures abbreviations use Fig. x and not fig x.

Figure 2. Caption  “  There is a strong correlation (R2=0.8) between “ should be changed as “ Correlation between early diagnosis and 5-year survival across cancer types”. The explanation should be described in the text.

Figure 2 caption  “ across “ Instead “avross”

The citations of figures in the text must be uniformed because, sometimes uses fig, sometimes figure, sometimes in uppercase, sometimes in lowercase

The citations of references in figure caprions must follow the manuscript format, that is, in numerical order and not by author, as happens in almost all the figures. So, for example in Figure 1 “Image from Griffen and Shockcor, 2004, … “ should be “Image from Griffen and Shockcor [number of reference].  The same for all other figures.

Please review references in all figure captions.

In the figures caption, it is intended to give a direct and objective information about the figure, what it represents. Any further explanation of figure content should be described in the text and not in figure captions. Also, it is not common star the figure caption with the reference, as for instance “Data from James 2010, illustrating the changes in response to hyperthermia ….” A caption deve traduzir o que representa a figura e depois então referenciar a sua proveniência. For exempla “ the changes in response to hyperthermia …. (adapted from [44] ”

In addition, most figure captions are very developed so the text irrelevant to the caption must pass into the body of the manuscript

Author Response

Dear Reviewer,

Thank you for the close read and recommended changes. 

I have rewritten the abstract, as it was in the original form from before I wrote the paper. It now more accurately reflects the manuscript and serves to help orient the reader.

Also, to aid readers, I have provided an overview of the approach taken in the final paragraph of the introduction section.

The captions and figure references reflect the recommended changes, further strengthening the manuscript.

Additionally, I have incorporated edits of the manuscript and generally improved readability.

Thank you for your suggestions, as I feel they have greatly improved the readability and therefore merit of the work.

Reviewer 2 Report

Taking care of health is one of the fundamental principles that people should follow. Early detection of emerging irregularities is crucial in this respect.

The manuscript presents a modern higly advanced technology of 7T MRI to diagnose very early forms of liver and pancreatic cancer.

The work is done very well, the possibilities of 7 T MRI are thoroughly discussed.  The manuscript is well organized and well written based on well-choosen literature.

This article will be of great interest to a wide range of readers.

Author Response

Thank you so much for your words of encouragement!

The manuscript has been revamped for readability -- including catching the typos.

Reviewer 3 Report

This survey makes a comprehensive analysis of the literature related to techniques of ultra high field proton imaging and phosphorus spectroscopy for improved outcomes and reduced collateral damage for liver and pancreatic cancers. The authors exploit imaging at 7 T, 7 T MRI and spectroscopy and the proof suggesting the potential of phosphorus spectroscopy for treatment monitoring. Also, authors discuss future applications of 7 T imaging and spectroscopy.

The survey provides a novel view of the research field, as it differs from existing surveys in terms of both extent and approach. The paper is well-written and organized, and the authors have conducted an impressive amount of work. The fact that it points towards such a large number of research fields and discusses a few works within its field to offer the reader a basic idea of research work and challenges and give valuable references, makes me lean towards its acceptance.

The quality of citations is good. They have referenced the major works in this field of research.

There are no critical technical flaws in the work.

Some typos across the document should be corrected. As an example, “spectrosopy” should be “spectroscopy”

Author Response

Thank you for your words of encouragement.

The manuscript has been fully revamped for readability -- the typos and run on sentences are no more! 

Reviewer 4 Report

The Authors suggest that since “…tumor tissues contain extreme higher levels of phospholipid metabolites compared to the background, even spectroscopic volumes containing less than 50%

of tumor can be detected and/or monitored. Phosphorus spectroscopy allows non-invasive pH measurements, indicating hypoxia, as a predictor of patients likely to benefit from treatments such as hyperthermia to increase efficacy of radiotherapy, chemotherapy, and/or immunotherapy. Once hypoxic regions are identified, phosphorus containing metabolites can be used for dosimetry during heat therapy. Preliminary research in breast cancer has confirmed the existence in vivo of a hitherto preclinical metabolic phenotype associated with aggressiveness, and identified the metabolic profile in a small lymph node.”

The message suggested by the Author is highly interesting and surely may offer a good possibility to be transferred to clinical needed, however the paper is quite confusing, and the reading not easy.

The English sentences are not always complete and/or clear, for example, in my opinion the first two sentences need more attention.

I think that the Authors should read carefully the ms and then resubmit it

Author Response

Thank you for your words of support and suggestions. Your review was very much in line with Reviewer 1. I hope that you both find that the revised manuscript is significantly improved in terms of readability.

Round 2

Reviewer 1 Report

The suggested corrections were done by the author in the revised version of the manuscript which improve significantly the scientific quality of the manuscript. Therefore, it achieve the scientific styandards to be published in Metabolites journal

Author Response

Thank you so much for your words of encouragement!